# Defining, Conceptualizing, and Measuring Organizational Resilience: A Multiple Case Study

**Ruijun Chen [1,\*], Yaping Xie [2] and Yingqi Liu [1,\*]**

1   School of Economics and Management, Beijing Jiaotong University, Beijing 100044, China
2   School of Economics and Management, Fuzhou University, Fuzhou 100044, China; 200720120@fzu.edu.cn
\*   Correspondence: 18113079@bjtu.edu.cn (R.C.); liuyq@bjtu.edu.cn (Y.L.)

**Abstract:** Organizational resilience is an important means of coping with crises. This concept has received much attention within both academia and industry. However, research on the definition and measurement of organizational resilience is still in the exploratory stage. To date, studies on organizational resilience have yielded mixed conclusions, which makes it difficult to provide specific recommendations for coping with crises. This paper uses an exploratory case study approach to explore the process of organizational resilience among six highly resilient companies: Southwest Airlines, Apple, Microsoft, Starbucks, Kyocera, and Lego. We employed grounded theory to distill the main characteristics of organizational resilience, to explore and validate its structural dimensions, and to develop a measurement scale for organizational resilience. Further, we conducted reliability and validity analysis, exploratory factor analysis, and validation factor analysis on the 526 valid data collected. Results show that organizational resilience includes five dimensions: capital resilience, strategic resilience, cultural resilience, relationship resilience, and learning resilience. The measurement scale has good reliability and validity, which better reflects the notion of organizational resilience. This study bridges the gaps in the existing literature on organizational resilience and its measurement scales, and provides a foundation for future research.

**Keywords:** organizational resilience; dimensions; multiple case study; scale development

## 1. Introduction

Today's business environment is increasingly complex and volatile [1]. With the globalization and the internationalization of business activities, crises seem to have become regular events in the development of organizations [2,3]. These "black swans" (small probability but high impact events) or "gray rhinos" (large probability and high impact potential crises) include the September 11 attacks, the 2008 financial crisis, the 2011 tsunami in Japan, the 2013 Ebola virus in Africa, and the COVID-19 pandemic in 2020. These events pose increasing challenges to the survival and development of organizations [4]. Therefore, how companies can manage risk and continue to grow during crises has become a key issue that must be addressed by decision makers. Why are some organizations better able to cope with adverse environmental conditions and survive in the face of a crisis? What kinds of processes lead to the adoption of new operation procedures? In essence, these are issues of organizational resilience [4–7]. Empirical and theoretical studies show that organizational resilience is the most direct factor explaining why companies can successfully overcome crises. This is because highly resilient companies have strong organizational resilience and they are able to overcome existential crises. There is growing evidence that resilient organizations have the ability to adapt to market changes and are more likely to remain relevant and responsive to market changes [8,9]. Some scholars have even linked resilience to the long-term development of companies [10]. For instance, Somers (2009) [11] defined organizational resilience as the ability of a firm to take measures in advance to cope with a crisis. Thus, the study of organizational resilience is of great importance, not only in

terms of providing new theoretical perspectives on crisis management, but also in terms of helping companies to grow through crises in practice.

After reviewing the research on organizational resilience, we found that this topic has received increasing attention in academic circles over the past 30 years [12–20]. Organizational resilience has been addressed in the fields of positive psychology [21], ecosystematics [22], engineering [23,24], and management [8,13]. Existing studies on organizational resilience have mainly focused on its definition and measurement, the factors influencing it, its mechanisms of operation, and its effects. Scholars have also looked at organizational resilience from the perspective of stakeholders [25–27], self-esteem [28,29], and leadership behavior [30–33]. However, most studies lack both a systematic theoretical exploration and a clear definition of organizational resilience [10,34]. Some scholars have conducted exploratory research on organizational resilience as an independent concept and proposed a measurement scale for organizational resilience [35–40]. However, scholars still do not have a unified opinion for the study of organizational resilience, and there are few in-depth evaluations of the dimensions of organizational resilience, which limits the scope of the research [39].

To address these limitations, this study provides an in-depth analysis of the concept of organizational resilience. We then conduct an exploratory multiple case study of six highly resilient companies, namely Southwest Airlines, Apple Inc., Microsoft, Starbucks, Kyocera, and Lego. We explore the dimensions of organizational resilience using grounded theory, develop measurement scales, and conduct empirical tests by combining theories and research results related to organizational resilience. Our study seeks to answer the following questions: What is organizational resilience? What are the processes involved in organizational resilience? How can companies in crisis achieve sustainable growth in spite of the crisis? We aim to help scholars better understand the meaning of organizational resilience and provide a reliable measurement basis for the empirical analysis of organizational resilience.

This paper contributes to the literature on organizational resilience in three ways. First, by reviewing existing research on organizational resilience, we found that it is a complex concept that is cross-level and multidimensional [39]. We clarify what organizational resilience is, providing an overarching definition and outlining how the specific process unfolds. In doing so, our study provides a systematic interpretation of the definition of organizational resilience in existing studies.

Second, scholars generally agree that organizational resilience is a multidimensional construct [27,41,42]. So far, scholars have built on the definition provided by McManus et al., (2008) [35] and they have added other aspects to the concept of organizational resilience. Based on organizational resilience theory, this study explores the structure of organizational resilience through qualitative research, selecting six highly resilient companies as research subjects. Using grounded theory to code our qualitative data, we divide organizational resilience into subcomponents, namely capital resilience, strategic resilience, cultural resilience, relational resilience, and learning resilience. This study makes creative use of qualitative research methods to make up for the shortcomings of existing studies on organizational resilience structures.

Third, we have developed a scale for measuring organizational resilience based on the studies of McManus et al., (2008) [35], Linnenluecke et al., (2012) [43], Godwin and Amah (2013) [44], and others. Thus far, scholars have constructed a measurement scale for organizational resilience from different perspectives and viewpoints based on the works of McManus et al., (2008) [35] and others [36,39,45]. However, fewer have used qualitative research methods to explore the structure of organizational resilience through the grounded theory research approach or developed a scale for measuring organizational resilience that has been accepted by other scholars. This study aims to provide a better measure organizational resilience and lays the foundation for further empirical studies.

The remainder of this paper is organized as follows. Section 2 reviews the literature on organizational resilience, summarizes its various definitions, and defines the meaning

of organizational resilience. Section 3 presents the research design and case analysis, describing the criteria for case selection, data collection, and analysis strategy. Section 4 presents the organizational resilience scale, including the initial measurement questions. Section 5 conducts the validity test for the scale, including the data collection, sample characteristics, and data analysis. Section 6 discusses the results obtained in this study, presents directions for future research, and concludes.

## 2. Literature Review and Theoretical Framework

The concept of resilience emerged in the late 1960s and early 1970s in the field of physics. It refers to the ability of a system to cope with change [46]. Beginning in the mid-1880s, the concept of resilience began to penetrate into the field of ecology, where it was first introduced by Holling (1973) [47] in the article entitled "Resilience and stability of ecological systems." In this article, the concept of resilience was introduced into the study of ecological environments, which refers to the ability of an ecosystem to recover to its previous state after being damaged. Later, Wildavsky (1988) [48] analyzed resilience in the context of organizational research. However, it was not until the late 1990s that the study of resilience within organizations gained popularity among scholars, who began to focus on post-disaster resilience research [44,49]. This includes research into events such as Hurricane Katrina, the September 11 attacks, as well as corporate resilience [50,51]. Moreover, there is a broader discussion on resilience in information systems [52], healthcare systems [13], and supply chains [53]. In recent years, organizational resilience has been well explored in the field of psychology. Researchers have studied psychologically well-adjusted children in high-risk environments, arguing that organizational resilience is the positive adaptive capacity that individuals show when experiencing adverse conditions [39].

In the organizational theory literature, resilience has been studied in the areas of disaster management, crisis management, and high-reliability organizations [54–57]. Nevertheless, scholars have not yet reached a unified conclusion regarding what constitutes organizational resilience [10]. As shown in Table 1, most existing studies interpret organizational resilience from the capability perspective, process perspective, functional perspective, and outcome perspective. We found that scholars with a dynamic view advocate exploring organizational resilience from a capability and process perspective, while scholars with a static view advocate exploring organizational resilience from an outcome and functional perspective. Scholars adopting a process perspective consider organizational resilience as a dynamic and progressive process exhibited by firms in response to crisis or adverse situations. Firms may adopt behaviors such as identity management, reintegration, improvisational coping, and emotional labor [6,58]. Scholars adopting the capability perspective consider organizational resilience as a dynamic and flexible organizational capability synthesized from predictive capability, survival capability, adaptive capability, coping capability, and learning capability that organizations exhibit in response to crises [1,20,59]. Scholars from an outcome perspective consider organizational resilience as the ability of organizations to remain in a positive adaptive state in the face of crises [60,61]. Scholars from a functional perspective view organizational resilience as a function of an organization's ability to adapt to dynamic and complex environments [35,62].

**Table 1.** Concept of organizational resilience.

| Perspectives | Representative Scholars | Concept Definition | Views |
|---|---|---|---|
| Capability Perspective | Duchek et al. (2020) | Organizational resilience is the ability to anticipate potential threats, to respond effectively to unexpected events, and to learn from these events, resulting in a dynamic capability designed to facilitate organizational change [1]. | Dynamic View |
| | Ma et al. (2018) | Organizational resilience is an organization's capability that enables organizations to survive, adapt, recover, and even thrive in the face of unexpected and catastrophic events as well as turbulent environments [59]. | |
| | Koronis and Ponis (2018) | Organizational resilience can be viewed not only as the ability to absorb or adapt to disturbances and changes, but also as the ability to recognize and adapt to unexpected changes [63]. | |
| | Annarelli and Nonino (2016) | Organizational resilience is the ability of an organization to face disruptions and unexpected events in advance due to shocks, both internal and external to the organization [4]. | |
| | Kim et al. (2016) | Organizational resilience is the ability of an organization to maintain its operations, and to adapt and recover from disasters [64]. | |
| | Ortiz-de-Mandojana and Bansal (2016) | Organizational resilience is the ability of an organization to be aware of disruptions and to respond proactively to unexpected events [65]. | |
| | Mafabi et al. (2015) | Organizational resilience is the ability of an organization to cope with change and to adjust its structure to prevent disruptions [66]. | |
| | Lengnick-Hall et al. (2011) | Organizational resilience is the ability of an organization to effectively absorb, develop context-specific responses to, and engage in change activities [20]. | |
| Process Perspective | Ishak and Williams (2018) | Organizational resilience is a dynamic structure of organizations that encompasses both typological and quantitative dimensions and covers processes such as reintegration, identity management, communication network building, emotional labor, and improvisational coping [58]. | |
| | McCarty et al. (2017) | Organizational resilience is an evolutionary process in which organizations respond to changes in the external environment by deploying resources [6]. | |
| | Lengnick-Hall and Beck (2009) | Organizational resilience competence is a process that develops from a combination of cognitive and behavioral competencies at the organizational level as well as knowledge, skills, attitudes, and behaviors at the individual level in contextual conditions [67]. | |
| | Allen and Toder (2004) | Organizational resilience refers to the process of tissue recovery from the damage caused by traumatic events [68]. | |
| Functional Perspective | Wicker et al. (2013) | Organizational resilience has been conceptualized as robust, redundant, adequate, and rapid functioning [62]. | Static View |
| | Mcmanus et al. (2008) | Organizational resilience is a function of an organization's awareness of the overall situation, its management of critical weaknesses, and its ability to adapt in a complex, dynamic, and interdependent environment [35]. | |
| Results Perspective | Sincorá et al. (2008) | Organizational resilience relates to how organizations survive and recover from unexpected events and chaotic changes. It includes the three dimensions of adaptation, anticipation, and recovery, which are closely linked [61]. | |
| | Gittell et al. (2006) | Organizational resilience is the result of both relational and financial reserves that enable organizations to maintain their relational reserves [69]. | |
| | Weick (1996) | Organizational resilience is the result of designing structures that are a source of resilience for collective perceived power [70]. | |

Our systematic review of the research on organizational resilience shows that the concept is applied in a number of fields such as ecology, psychology, and economics. In summary, organizational resilience contains three main essential elements. First, the organization operates in a dynamic environment. Second, the organization responds to the crisis by reconfiguring organizational resources, reshaping organizational relationships, and optimizing organizational processes in an adverse situation. Third, the organization reaches recovery and achieves growth. Therefore, we regard organizational resilience as the ability of an organization to reconfigure organizational resources, optimize organizational processes, reshape organizational relationships in a crisis, recover quickly from the crisis, and use the crisis to achieve counter-trend growth. Organizational resilience emphasizes the ability of companies to not only get out of a difficult situation, but also to drive growth in a crisis.

Although scholars have interpreted the concept of organizational resilience from different perspectives, there is no universally accepted standard for the study of the structure and its measurement. As shown in Table 2, most of the existing studies can be divided into two-factor, three-factor, and four-factor structures. McManus et al. (2008) [35] proposed that organizational resilience encompasses planning capacity and adaptive capacity. Based on an analysis of 10 firms in the New Zealand region, the authors developed 13 resilience indicators to measure organizational resilience. Later, scholars such as Godwin and Amah (2013) [44] and Umoh et al. (2014) [41] incorporated organizational learning into organizational resilience. Godwin and Amah (2013) developed separate 15-item scales measuring organizational resilience using 128 employees from 34 manufacturing firms in Nigeria. Borekci et al. (2014) [71] focused on organizational structure and organizational capacity. The authors conducted an analysis of 109 service firms in Turkey. They developed a 15-item measurement scale and suggested that organizational resilience encompasses structural dependence, organizational capacity, and process continuity. Kantur and Say (2015) [39] argued that organizational resilience includes robustness, agility, and integrity, and developed a 10-item scale for measuring organizational resilience. In recent years, scholars have generally considered organizational resilience as a four-dimensional construct, with some arguing that it includes robustness, redundancy, adequacy, and rapidity [62,72–74]. According to Richtnér and Löfsten (2014) [75], organizational resilience includes structural, cognitive, relational, and emotional competencies. They came to their conclusions through an analysis of 329 technology-based companies in Sweden, and developed a 14-item measurement scale.

Overall, the study of organizational resilience has been widely discussed in the fields of organizational behavior and strategic management. While qualitative and quantitative research continues to evolve, quantitative research on organizational resilience has developed more slowly. This is mainly due to the fact that scholars have different perceptions of organizational resilience, and there is still a lack of a uniform scale for measuring organizational resilience [39,40]. This study seeks to address this gap by constructing a unified definition of organizational resilience through an exploratory case study approach and then develop a scale to measure organizational resilience.

**Table 2.** Dimensional components and related measures of organizational resilience.

| Structure Type | Research Sample | Dimensional Composition (Number of Items) | Source of Measurement Items | Representative Literature |
|---|---|---|---|---|
| Two factors | 10 companies in New Zealand | Planning ability (5) Adaptability (8) | [35] | [76] |
| | 84 tourism companies in New Zealand | Adaptability (7) Planning (6) | [36] | [77] |
| | —— | Organizational awareness (5) Adaptability (6) | [36,78] | [79] |

**Table 2.** *Cont.*

| Structure Type | Research Sample | Dimensional Composition (Number of Items) | Source of Measurement Items | Representative Literature |
|---|---|---|---|---|
| Three factors | 128 employees in 34 manufacturing companies in Nigeria | Organizational Adaptation (5) Organizational resources (5) Organizational learning (5) | [37,80] | [44] |
| | 31 manufacturing industries in Nigeria | Organizational learning ability (5) Adaptability (5) Dynamic ability (5) | [38,81,82] | [41] |
| | 109 service companies in Turkey | Structural dependence (3) Organizational capacity (3) Process continuity (3) | [83] | [71] |
| | 188 companies in Turkey | Robustness (4) Agility (3) Integrity (3) | [55] | [39] |
| | 10 organizations in New Zealand | Situational Awareness (5) Critical vulnerability management (5) Adaptability (5) | [84] | [85] |
| Four factors | 739 community clubs in Australia | Robustness (5) Redundancy (5) Adequacy (6) Speediness (5) | [40,72] | [62] |
| | 329 technology-based companies in Sweden | Structural ability (4) Cognitive ability (3) Relational ability (3) Emotional ability (4) | [40] | [75] |
| | 7 organizations in Turkey | Expected competencies (5) Adaptive culture (4) Network competency (6) Organizational learning (4) | [86–88] | [89] |

## 3. Research Design and Case Study Analysis

### 3.1. Case Selection

This study utilized an exploratory case study method. We selected six highly resilient companies, based on their typicality and the ease of access to information: Southwest Airlines, Apple, Microsoft, Starbucks, Kyocera, and Lego (see Table 3). First, two criteria were set for case selection. We selected companies that have been in existence for more than 40 years, and that had suffered a major crisis but emerged successfully and achieved sustained growth. These are highly resilient companies from Japan, the United States, and Denmark, respectively. These six companies have been in existence for more than 40 years. The oldest company is Lego, which was founded in 1932 and has 88 years of history. The youngest company is Apple, which was founded in 1976, with 44 years of history. There is no doubt that these six companies have experienced many crises over the past few decades, and they have all managed to survive and they have experienced growth against all odds. Although these six companies encountered different types of crises (some arising from the external environment and some from the lack of internal strategy), the resilience of these companies is worth studying.

Second, we selected the cases based on the ease of access to information. The annual reports of highly resilient companies contain important analytical material in the form of letters from the CEO or Chairman to investors. Each letter details the issues faced by the company in a respective year and the major initiatives the company has taken. In times of crisis, the annual report analyzes the reasons why the company survived the crisis. We also analyzed media reports and management commentaries on the six companies that were published in the Harvard Business Review, the Wall Street Journal, Business Week, and The Economist. We also studied books that were published about the six companies. In the process of screening the textual data, we adopted a problem-oriented approach to record and organize the events related to these six companies and the crises they faced. The

database contained over 200 key events that occurred in these six companies in response to a crisis, which provides a strong basis for this study.

**Table 3.** Information on the companies selected.

| Company | Country | Founding Date | Case Type | Case Features |
|---|---|---|---|---|
| Southwest Airlines | United States | 1971 | Core Case | Southwest Airlines has experienced four major crises: the first was from 1979–1985, the second from 1990–1997, the third was from 2001–2007, and the fourth was from 2008–2015. From 1973–2019, the company has been consistently profitable for 47 years. |
| Apple | United States | 1976 | Comparative case | Apple was on the verge of collapse in 1996, then Steve Jobs stepped in and reinvented the company's business model in 1997. Apple became the first high-tech company to reach $1 trillion in market capitalization. |
| Kyocera | Japan | 1959 | Comparative case | Kyocera has weathered several crises over the past 60 years, including the financial crisis, the Internet bubble, and the Great Earthquake. To date, Kazuo Inamori has led the company, which has achieved sustained growth for 59 years. |
| Starbucks | United States | 1971 | Comparative case | When Starbucks was on the verge of bankruptcy in 2008, Howard Schultz became CEO and led the company out of the crisis by reinventing the business model and achieving sustained growth. |
| Lego | Denmark | 1932 | Comparative case | Beginning in 1997, Lego faced an eight-year financial crisis and was on the verge of bankruptcy. In 2004, Jørgen Wiig-Knustop took over as CEO, leading the company out of the crisis and reinventing Lego's business model to achieve growth. |
| Microsoft | United States | 1975 | Comparative case | In 2014, Microsoft was in a "strategic crisis" in the mobile Internet space and consumer hardware industry. Satya Nadella became Microsoft's CEO, leading Microsoft out of the crisis by reinventing its business model to reach a market capitalization of $1 trillion. |

### 3.2. Data Collection and Analysis

We collected textual data with information on the six companies from company annual reports, published books, industry information, and the Harvard Business Report. To ensure the validity and completeness of the data, this study triangulated the information from different data sources. Specifically, the information collected was verified through multiple channels and in different ways. For example, we verified the authenticity of the information by communicating with the relevant managers of the companies, in order to enhance the robustness of the conclusions. To ensure the accuracy of data analysis, we also invited teachers and Ph.D. students with experience in conducting case studies to assist as we conducted our research.

### 3.3. Data Analysis

This study follows the research procedure outlined in procedural rooting theory to sort and code the obtained data. Rooting theory is a top-down research method that constructs theories in an inductive form [90]. Compared to constructive rooting theory and classical rooting theory, the procedural rooting method is clearly structured and simple to operate. This not only provides rigorous guidelines for researchers, but also offers

concise operational steps. For example, the research procedure for programmed grounded theory includes three processes: open coding, spindle coding, and selective coding [91]. In order to ensure the accuracy and rigor of the coding, a coding team consisting of three doctoral students, two master's students, and one professor was established. Each member independently coded the textual information and subsequently compared and adjusted the codebook to improve the reliability of the study.

### 3.3.1. Open Coding

Open coding is the process of interpreting the collected data, exploring the meaning behind the data, and using conceptualization and categorization to condense the intrinsic essence of the data, with the ultimate goal of achieving data aggregation. The open coding process requires the researcher to approach the data with a critical perspective and to be problem-oriented so that the information can be captured accurately. Table 4 shows the open coding process.

**Table 4.** Open coding results.

| Collated Information | Conceptualization | Categorization |
|---|---|---|
| Our goal is to design a capital structure that leverages all capital to maximize returns for shareholders over the long term. Lego's problems are rooted in a disconnect between internal strategic upgrading and operational capabilities, and, furthermore, in a mismatch between strategic objectives and capital structure. | Capital Structure Design Capital Structure Matching | Capital Structure |
| Throughout the oil crisis, Southwest maintained a solid cash flow, thus, ensuring its own sustainable growth in the face of adversity. From 1980 to 1983, Southwest Airlines issued 3.6 million shares of common stock on the open market for four consecutive years, greatly enhancing the company's cash flow. At the onset of the crisis caused by the 9-11 crisis, the company had $1 billion in cash on hand, which enabled it to respond to a crisis that depleted their cash flow. | Stable Cash Flow Balanced Equity Financing Cash Sufficient | Cash Storage |
| The cash flow was first used to pay off the company's debt, which rapidly fell from $72 million in 1979 to $15 million in 1980. The losses continued to accumulate in 2004, reaching DKK 1.931 billion, and Lego was in a serious existential crisis and on the verge of bankruptcy. | Priority Debt Service Debt Crisis | Debt Service |
| Southwest Airlines continued to improve the company's solvency. Even in 1979, 1980, and 1981, the most difficult years of the crisis, Southwest's current ratio reached 1.64:1, 1.53:1, and 1.23:1 | Solvency Higher Current Ratio | Debt service Capacity |
| Why does Southwest Airlines stand out from the competition? The answer is simple and clear: the price of their product. They believe that in the market for short-haul flights of 500 miles or less, the private car is the primary competitor to the airplane. ... This requires them to continuously optimize the cost structure of their product and lower the price of their product so that passengers feel that flying is more punctual and cheaper than using a private car. In short, the only way to win is to offer an attractive product at a competitive price. | Identify Competitors Optimize Cost Structure Reduce Product Prices | Product Features |
| Continue to improve aircraft utilization efficiency (over 10 h per day) and accelerate aircraft return times (10 min for ground boarding). Continue to use standardized and uniform aircraft types, primarily Boeing 737s. No cost-increasing catering services, which is unnecessary on short-haul routes. Focus on serving passengers (no cargo or mail service) and carrying smaller baggage that is more profitable and less expensive to handle. | High Efficiency Simple Operations Focus on Passengers | Operation Strategy |
| Pan Am reduced tickets for business flights from Dallas to Houston to $13, compared to Southwest's cost of about $13. Pan Am's pricing strategy took advantage of the first-mover advantage and drove Southwest to unprofitability. | Low Price Price First | Price Conflict |
| Iranian oil exports were completely halted from December 1978 to March 1979 due to dramatic domestic political changes. The U.S. economy was severely affected and its domestic gasoline prices rose from $0.65 in 1978 to $1.35 in 1981, leading to a recession in the U.S. economy. | Oil crisis Economic recession | Survival Crisis |

**Table 4.** *Cont.*

| Collated Information | Conceptualization | Categorization |
|---|---|---|
| "I have been writing to praise the dedication, enthusiasm, and outstanding achievements of Southwest Airlines employees.<br>With tears in my eyes, I thank all employees from the bottom of my heart for their support and care for the company.<br>There were many opportunities to lay off employees and make the company more profitable, but I always thought that was short-sighted." | Commending Employees<br>Appreciate Employees<br>Retaining Employees | Emotional Connection |
| This year, Southwest launched the "Fuel from the heart" program, voluntarily reducing wages to allow the company to purchase fuel.<br>This reaffirms that Southwest is not just a business company, we are a family that cares about each other.<br>They are committed to being a low-cost leader, to achieving superior financial performance and to protecting the safety of our employees and shareholders. | Helping Companies<br>Caring for Each Other<br>Safeguarding Stakeholders | Reciprocal Relationship |
| Southwest's primary goal is to provide safe, compassionate, caring and courteous service to its passengers.<br>Southwest has made "service excellence" a core monitoring indicator of its operations and has not compromised its service standards, even in the most difficult of times.<br>Southwest Airlines provides a happy, welcoming service on its flights, allowing passengers to enjoy a relaxing journey. | Providing Quality Services<br>Excellent Service<br>Warm Service | Customer Service |
| Southwest Airlines increases customer stickiness through more personal and warmer service.<br>Not laying off employees fosters loyalty and a sense of security and trust. | Customer Stickiness<br>Employee Loyalty | Relationship Enhancement |
| "Our employees were immersed in great grief, but they buried their grief deep in their hearts, and with tears in their eyes, they quickly returned to their respective posts, and they re-planned their flights as quickly as possible to ensure the normal operation of the route.<br>Always follow the right path, which gives us endless strength in the crisis and makes us insist on the right path and the right thing to do when we encounter any difficulties.<br>This is because our employees have the spirit of hard work and dedication, as well as a good sense of humor and loyalty to the organization." | Work Commitment<br>Road Commitment<br>Organizational Loyalty | Employee Commitment |
| The company likes to hire people with diverse backgrounds and places great emphasis on examining the positive and optimistic spirit of the candidates.<br>Southwest emphasizes cultural diversity and teamwork.<br>Due to Southwest's innovation in reducing the cost of short-haul flights, its miles flown are now positively correlated with seat-mile costs. | Optimism<br>Team spirit<br>Innovative spirit | Spiritual Shaping |
| Herb Kelleher and his team fought so hard that not only were they not beaten down, but they grew stronger and stronger and achieved continued growth in the face of adversity.<br>The employees were made to realize that the airline industry is in a highly competitive, dynamic, and risky environment and that to survive, one must have the spirit of struggle and love to fight to win.<br>The spirit of caring and joy has continued from the beginning of Southwest Airlines' business to today. | The more you fight, the stronger you are<br>Passion for Struggle<br>Care and Happiness | Rigid and Flexible |
| Through the profit sharing program, Southwest Airlines and all employees form a community of interest, and when the company's profits increase, employees can receive a share of the profits.<br>Kyocera's culture of supreme goodness also promotes care among people. By caring for others and extending kindness to them, such kindness will reincarnate and eventually benefit oneself, thus, forming a community of destiny that helps each other. | Community of Interest<br>Community of Fate | Community Sense |
| Southwest Airlines' spirit of joy and caring, coupled with the warrior spirit, acts as an emotional catalyst, mobilizing positive energy and positive emotions among employees.<br>For any organization, morale is an important expression of individual emotions. High morale is a positive and positive emotion that increases the likelihood and speed of an organization overcoming a crisis.<br>A culture of excellence makes employees realize that they can contribute to people by realizing the dream of "letting customers fly free," which enhances the individual's perception of the meaning of life and the sense of accomplishment that comes from work. | Mobilize emotions<br>Raise morale<br>Perception of the meaning of work | Emotional Regulation |

**Table 4.** *Cont.*

| Collated Information | Conceptualization | Categorization |
|---|---|---|
| "Southwest Airlines is an exact copy of PSA, literally a copy of PSA, so to speak." Lamar Muse, in an interview, never hid Southwest's early copying of PSA. Kyle embarked on a radical transformation, upgrading the company's positioning from building toys to creative and inspiring quality toys. Southwest's leaders were acutely aware that as a business organization, the company must prosper with its employees. | Model Awareness Positioning Awareness Situational awareness | Positive Awareness |
| If we trust only when we are not let down, love only when we have something to give in return, and learn only when we have something to learn, then we have abandoned the essential characteristics of being human. When learning from the experience of others, companies should set a safety standard for gearing based on their own operational characteristics and consider it as a strict financial discipline to be adhered to for a long time. | Intrinsic Characteristics Operating Characteristics | Behavioral Characteristics |
| Herb Kelleher, the founder of Southwest Airlines, decided to replicate the PSA model exactly, and he systematically studied and learned PSA's operating model and service model. When the country and the company were in an emergency situation, they had no complaints and quickly learned the new safety regulations and operating procedures established by the federal government. Many leaders who have copied the Southwest model have made a fundamental mistake in their strategic thinking by failing to understand the essence of the Southwest model in a systematic and balanced way, learning only superficial management techniques or marketing strategies. | Systematic Learning Fast Learning Deep Learning | Learning Capabilities |

It is important to note that different sample profiles may respond to different aspects, which means that a category or concept may come from one sample, while others play a complementary and supportive role. The main goal of this study is to find the concepts and categories that best describe, categorize, and compare the data. A total of 107 concepts and 20 categories were obtained in this study.

### 3.3.2. Axial Coding

Axial coding is the process of further sorting and generalizing the categories established through open coding in order to discover the relationships between the categories. We used the process referred to as main axis coding, which is mainly based on the following model: "causal condition—phenomenon—action strategy—outcome" [92]. We clarified the relationships between the categories obtained in the open coding and form larger classes, thus, forming the master category [92]. After generalization and refinement, we finally obtained five categories, namely capital resilience, strategic resilience, cultural resilience, relationship resilience, and learning resilience (see Table 5).

**Table 5.** Axial coding.

| Logic Main Line | | | | Main Category |
|---|---|---|---|---|
| Causal Conditions | Phenomenon | Action Strategy | Results | |
| Capital Structure | Cash Reserve | Debt Service | Debt Service | Capital Resilience |
| Survival Crisis | Price Conflict | Operation Strategy | Product Features | Strategic Resilience |
| Employee Commitment | Spiritual Shaping | Rigid and Flexible | Community Sense | Cultural Resilience |
| Emotional Connection | Reciprocal Relationship | Customer Service | Relationship Enhancement | Relationship Resilience |
| Emotional Regulation | Behavioral Characteristics | Positive Awareness | Learning Ability | Learning Resilience |

### 3.3.3. Selective Coding

Selective coding is based on the axial coding, and a more complete structure emerges by analyzing around the core categories so as to discover the core categories that can unify

the other categories. Based on the interaction of resilience theory and data, the four main categories of capital resilience, strategic resilience, cultural resilience, relationship resilience, and learning resilience were extracted. Moreover, the core category of organizational resilience was further specified, which can unify all the categories of this study. The selective coding process is shown in Figure 1.

| Category | Category | Category | Category | Category |
|---|---|---|---|---|
| Capital Structure | Survival crisis | Employee Commitment | Emotional Connection | Emotional Regulation |
| Cash Reserve | Price Conflict | Spiritual Shaping | Reciprocal Relationship | Behavioral Characteristics |
| Debt Service | Operation Strategy | Rigid and Flexible | Customer Service | Positive Awareness |
| Debt Service | Product Features | Community Sense | Relationship Enhancement | Learning Ability |
| Capital Resilience | Strategic Resilience | Cultural Resilience | Relationship Resilience | Learning Resilience |
| Main Category | Main Category | Main Category | Main Category | Main Category |

Core Categories
Organizational Resilience

**Figure 1.** Selective coding process.

A main finding of this study is that strategic resilience helped companies consistently identify and eliminate adverse factors that could weaken a company's core business capabilities. Capital resilience helps companies balance their organizations' capital structure and prepare for crises before they occur. Relationship resilience helped companies build mutually beneficial relationships between employers and employees, employees and companies, and companies and customers to help them withstand crises. Cultural resilience was the shaping of employees through corporate culture, encouraging them to have a long-term commitment toward the organization. Learning resilience is the ability of an organization to cope with the stresses it faces in the learning process, and the ability of an organization to learn various lessons for in response to a crisis. Together, these five factors formed the organizational resilience that helped companies navigate through crises and achieve sustained growth.

## 4. Developing the Organizational Resilience Scale

### 4.1. Scale Development Process

First, following existing studies, we used programmed root theory to conduct multiple case studies. We obtained five dimensions of organizational resilience: capital resilience, strategic resilience, cultural resilience, relationship resilience, and learning resilience. Second, we developed measurement items that are consistent with organizational resilience by drawing on existing measurement scales. Third, in order to analyze organizational resilience more clearly and to modify the measurement items, we conducted semi-structured interviews with the heads of the six companies who had experienced crises. During the interviews, we asked them what factors could help companies survive crises, or what organizational actions could help companies resist crises. Finally, we conducted a pre-study of the developed scale and further revised the scale to form a formal measure of organizational resilience.

## 4.2. Preparation of Initial Measurement Scale

In this study, the initial measurement scale was developed based on semi-structured interviews and combined with existing scales. Our measure of capital resilience was designed based on the studies of Valikangas (2010) [37] and Marsick et al. (2002) [80]. We created a total of seven questions in Likert-scale format, including "Our company has good cash flow", "We reserve cash according to our corporate strategy and competitive model", and "We have a sound capital structure".

The measure of strategic resilience was designed based on the study by Davenport and Cronin (2000) [93]. We created a total of six questions, including "Our company is able to focus on its core business", "Our company is able to identify adverse factors in its development in a timely manner", and "We pursue a robust strategic growth model".

The measure of cultural resilience was designed based on the studies by Meen and Keough (1992) [94] as well as Denison and Mishra (1995) [95]. It comprised a total of six questions, such as "Our corporate culture aims to develop a sense of community among employees", "Our corporate culture fosters a sense of cooperation among employees", and "Our corporate culture stimulates morale and spirit among employees".

The measurement for relational resilience was mainly designed based on the studies by Shore et al. (1990) [96] and by Vogus and Sutcliffe (2007) [40]. We created a total of six questions, such as "We are able to create unique value for our customers", "We aim for mutual prosperity between the company and our stakeholders", and "We have a good reciprocal relationship with our employees".

The measurement for cultural resilience was mainly designed based on the studies by Costanza et al. (2016) [87] and Ramón and Koller (2016) [88]. We created a total of six questions, such as "We will choose learning objects according to the characteristics of our own enterprise ", "We will choose the better enterprises to learn from", and "We will have a deep knowledge of our own situation in time". The questions are shown in Table 6 below.

**Table 6.** Initial measurement scale of organizational resilience.

| Dimension | Related Concepts | Scale Items | Source |
|---|---|---|---|
| Capital Resilience | The ability of a business to operate normally and to recapitalize against risk in a crisis. | Our business has good cash flow | [37,80] |
| | | We will base our cash reserves on our corporate strategy and competitive model | |
| | | We have a solid capital structure | |
| | | We have multiple sources of financing | |
| | | We have low capital leverage | |
| | | We will make profit maximization the ultimate goal of our business | |
| | | We have high capital utilization efficiency | |
| Strategic Resilience | Companies are able to maintain strategic consistency over time, helping them to identify and eliminate disadvantages and to be able to choose the right growth model. | Our company is able to focus on its core business | [93] |
| | | Our company is able to identify unfavorable factors in development in a timely manner | |
| | | We pursue a robust strategic growth model | |
| | | We were able to clarify our strategic positioning | |
| | | We are able to balance endogenous and exogenous growth patterns | |
| | | We are able to match strategic objectives and operational capabilities very well | |
| Cultural Resilience | Corporate culture shapes the entrepreneurial spirit of employees and their commitment to the organization. | Our corporate culture is designed to foster a sense of community among our employees | [94,95] |
| | | Our corporate culture fosters a sense of cooperation among our employees | |
| | | Our corporate culture inspires employee morale and spirit | |
| | | Our corporate culture inspires employees to strive for excellence | |
| | | Our corporate culture reflects the care and love for our employees | |
| | | Our corporate culture fosters a sense of organizational commitment | |

**Table 6.** *Cont.*

| Dimension | Related Concepts | Scale Items | Source |
|---|---|---|---|
| Relationship Resilience | Reciprocal relationship between business and stakeholders. | We can create unique value for our customers | [40,96] |
| | | We are able to think about our customers | |
| | | We aim for shared prosperity between companies and stakeholders | |
| | | We have a good reciprocal relationship with our employees | |
| | | We have a good relationship with our investors | |
| | | We are able to fully listen to the advice of our investors | |
| Learning Resilience | The ability of companies to cope with the pressures and challenges in learning. | We will choose the learning target according to the characteristics of our own company | [87,88] |
| | | We will choose the better companies to study | |
| | | We will have a deep awareness of our situation in time | |
| | | We will make timely adjustments to our positioning | |
| | | We will be interested in adjusting our emotions to get into the study state more quickly | |
| | | We will learn more about other experiences to help companies cope with the crisis | |

Source: Based on relevant literature and interviews with relevant business leaders.

## 5. Validity Test for the Scale

### 5.1. Data Collection and Sample Characteristics

#### 5.1.1. Data Collection

This study was conducted over a total of five months, from July 2020 to the end of November 2020. The questionnaire was administered in Beijing, Fujian, Zhejiang, and Shanghai, where there are clusters of small- and medium-sized enterprises. The questionnaire included questions about participant demographics and about organizational resilience. The latter set of questions were designed using a 7-point Likert scale, with "1" representing complete disagreement and "7" representing strong agreement. We selected small- and medium-sized enterprises that operated in clusters for several reasons. First, these enterprises are considered resilient family businesses. Second, these enterprises better reflect the characteristics of resilience. Third, given the difficulties involved in collecting data on enterprises nationwide, the regions selected for this study comprise more enterprises than in other areas of China. Fourth, considering that the Chinese economy is dominated by small and medium-sized enterprises, it made more sense to select small and medium-sized enterprises, as this represents the situation in China more accurately.

We obtained data for this study through three main methods. First, we relied on the resources of the local Institute of Entrepreneurship and Enterprise Development to conduct questionnaire surveys in Beijing, Fujian, Zhejiang, and Shanghai, and data were collected through mail and electronic questionnaires. Second, we conducted research on eligible companies in the four regions of Beijing, Fujian, Zhejiang, and Shanghai, collecting data mainly through on-site questionnaires.

#### 5.1.2. Sample Characteristics

In this study, 900 questionnaires were distributed in four regions, and a total of 723 questionnaires were collected. By eliminating invalid questionnaires that were incomplete and that were filled out randomly, 526 valid questionnaires were finally obtained. The sample characteristics of this study are shown in Table 7 below.

**Table 7.** Sample characteristics.

| | Options | Frequency | Percentage | | Options | Frequency | Percentage |
|---|---|---|---|---|---|---|---|
| Sex | Male | 275 | 52.3% | | Less than 1 year | 69 | 13.1% |
| | Female | 251 | 47.7% | | 1–5 years | 158 | 30.0% |
| Age | 25 years old and below | 125 | 23.8% | Years of work | 6–10 years | 93 | 17.7% |
| | 26–35 years old | 238 | 45.2% | | 11–15 years | 87 | 16.5% |
| | 36–45 years old | 138 | 26.2% | | 15–20 years | 78 | 14.8% |
| | 46–55 years old | 18 | 3.4% | | Over 20 years | 41 | 7.8% |
| | 55 years old and above | 7 | 1.3% | | General staff | 197 | 37.5% |
| Education level | Junior high school and below | 14 | 2.7% | | Basic manager/junior | 93 | 17.7% |
| | High school/junior high school | 62 | 11.8% | Position Level | Middle management/Middle level | 146 | 27.8% |
| | College/Bachelor's | 324 | 61.6% | | Senior management/Senior level | 83 | 15.8% |
| | Master's | 106 | 20.2% | | Other | 7 | 1.3% |
| | Ph.D. | 20 | 3.8% | | | | |

*5.2. Data Analysis*

Based on the scale development process, we first conducted an internal consistency analysis of the scale, i.e., the degree of consistency of multiple measures measuring the same concept; secondly, this study examined the structural validity, content validity, criterion validity, and face validity of the scale to ensure the validity of the scale; thirdly, we conducted an exploratory factor analysis (EFA) on the obtained scale to determine the optimal structure of the measurement scale and to judge the conformity of the measurement items contained in the scale with the corresponding concepts; finally, we conducted validation factor analysis (CFA) on the basis of exploratory factors to explore the relationship between theoretically delineated factors and the designed measurement items.

5.2.1. Internal Consistency Analysis

Drawing on Landers' (2015) [97] requirements for internal consistency testing, this study applied the Intraclass Correlation Coefficient (ICC) measure to test the internal consistency of the scale of organizational resilience, as shown in Table 8 below, which shows that the internal consistency of the scale was 0.911, indicating that the scale has good internal consistency.

**Table 8.** Intraclass Correlation Coefficient (ICC) for Inter-Rater Assessment.

| | Intraclass Correlation [b] | 95% Confidence Interval | | F Test with True Value 0 | | | |
|---|---|---|---|---|---|---|---|
| | | Lower Bound | Upper Bound | Value | df1 | df2 | Sig |
| Single Measures | 0.222 [a] | 0.180 | 0.266 | 6.955 | 525 | 7190 | 0.000 |
| Average Measures | 0.911 | 0.867 | 0.945 | 6.955 | 525 | 7190 | 0.000 |

Note: [a] The estimator is the same, whether the interaction effect is present or not. [b] Type A intraclass correlation coefficients using an absolute agreement definition.

5.2.2. Validity Analysis

Structural validity can demonstrate the reliability of the scale structure, but reliability tests do not represent validity. Chan and Idris (2017) [98] suggested that factor analysis

would be an appropriate statistical measurement technique for structural validity, with measures of Kaiser-Meyer-Olkin (KMO) and Bartlett's Test of Sphericity. Therefore, Table 9 below explains the structural validity of the scale.

**Table 9.** KMO and Bartlett's test for validity.

| Kaiser-Meyer-Olkin Measure of Sampling Adequacy | | 0.881 |
|---|---|---|
| Bartlett's Test of Sphericity | Approx. Chi-Square | 6862.277 |
| | df | 190 |
| | Sig. | 0.000 |

Table 9 explained that the KMO value is 0.881, which can be conclude by meritorious, and this indicator is greater than the minimum standard 0.6. Thus, it indicates that the empirical measurement effectively tests the true meaning of the concept under consideration. Moreover, the approximate chi-square of Bartlett's sphere test is 6862.277 with 190 degrees of freedom. *p*-value is 0.000, which is less than 0.01. Therefore, there is sufficient evidence to conclude that the questionnaire is valid at the 99% significance level.

Content validity analysis was conducted using a 4-point expert scale, with scores from 1 to 4 representing "not relevant", "weakly relevant", "strongly relevant", and "very relevant". In this study, the content validity index (I-CVI: the number of experts who rated each entry as 3 or 4 divided by the total number) was calculated based on expert ratings, and the average content validity (S-CVI/Ave) was the mean of the I-CVI of all entries. An I-CVI value greater than 0.78 and an S-CVI value greater than 0.9 indicated good internal homogeneity [99]. In this study, six experts in the field were invited to rate the entries of the scale, and the results of both I-CVI and S-CVI were 1. Therefore, the content validity of this scale is good.

To evaluate criterion validity, it could calculate the correlation between the results of the measurement and the results of the criterion measurement. This study generates the new variable for Total to represent the sum of value for the whole question and if there is a high correlation among the questions, it gives a good indication that the test is measuring what it intends to measure. For this indicator, the obtained value needs to compare with the critical value for Pearson correlation coefficient and the rule is that the obtained value should be greater than the critical value. For instance, the obtained value for the first item is 0.486. The sample size is $n = 526$, so the number of degrees of freedom is $df = n - 2 = 526 - 2 = 524$. Then the corresponding critical correlation value r_c for a significance level of $\alpha = 0.05$, for a two-tailed test is 0.085. Thus, the obtained value is greater than the critical value and this would generate the significant result for criterion validity. It can be seen that the scale has good criterion validity, as shown in Table 10 below.

**Table 10.** Test the validity using Pearson correlation coefficient.

| | Pearson Correlation | Sig. (2-Tailed) |
|---|---|---|
| Our business has good cash flow | 0.486 ** | 0 |
| We have multiple sources of financing | 0.750 ** | 0 |
| We have low capital leverage | 0.701 ** | 0 |
| We will make profit maximization the ultimate goal of our business | 0.757 ** | 0 |
| Our company is able to focus on its core business | 0.783 ** | 0 |
| We pursue a robust strategic growth model | 0.647 ** | 0 |
| We were able to clarify our strategic positioning | 0.607 ** | 0 |
| We are able to balance endogenous and exogenous growth patterns | 0.634 ** | 0 |
| Our corporate culture is designed to foster a sense of community among our employees | 0.607 ** | 0 |

**Table 10.** *Cont.*

|  | Pearson Correlation | Sig. (2-Tailed) |
|---|---|---|
| Our corporate culture inspires employee morale and spirit | 0.582 ** | 0 |
| Our corporate culture reflects the care and love for our employees | 0.573 ** | 0 |
| Our corporate culture fosters a sense of organizational commitment | 0.502 ** | 0 |
| We can create unique value for our customers | 0.458 ** | 0 |
| We aim for shared prosperity between companies and stakeholders | 0.456 ** | 0 |
| We have a good reciprocal relationship with our employees | 0.424 ** | 0 |
| We have a good relationship with our investors | 0.534 ** | 0 |
| We will choose the learning target according to the characteristics of our own company | 0.621 ** | 0 |
| We will have a deep awareness of our situation in time | 0.674 ** | 0 |
| We will be interested in adjusting our emotions to get into the study state more quickly | 0.601 ** | 0 |
| We will learn more about other experiences to help companies cope with the crisis | 0.650 ** | 0 |

Note: ** Correlation is significant at the 0.01 level (2-tailed).

Face validity refers to whether the questionnaire, in its surface form and content, is perceived as a valid questionnaire in the hands of the subjects [100], and Harrison and Wills (1983) [100] states that the only way to know the surface validity is through questionnaires or informal questioning of teachers and students. Therefore, in this study, two professors and five doctoral students in the field were invited to evaluate the face validity of this scale based on the preliminary research, and finally the experts agreed that this scale has good face validity.

5.2.3. Exploratory Factor Analysis

This study used SPSS 26.0 to conduct exploratory factor analysis on the 31 measurement items of the organizational resilience scale, and then to determine the optimal structure of the scale. We found that the KMO coefficient of the organizational resilience measurement scale was 0.881, which was within the valid range. The factor significance level of Bartlett's sphericity test was less than 0.001, which rejected the hypothesis that the correlation matrix was a unitary array, and, thus, the samples collected by this questionnaire were eligible for doing factor analysis. Next, we conducted an exploratory factor analysis using principal component analysis and found that a total of five factors were extracted. In addition, the questions with factor loadings of less than 0.4 were eliminated. These included "We reserve cash based on our corporate strategy and competitive model" and "We have a sound capital structure".

The final measurement scale was obtained by optimizing the scale and removing the unqualified measurement questions. As shown in Table 11, the factor loadings of all five factors were above 0.6, and the cumulative variance contribution rate reached 81.059%, indicating that the organizational resilience scale has a good factor structure. Based on programmed rooting theory and on the literature on organizational resilience, this study corresponds these five factors to the five dimensions of organizational resilience. These are: (1) capital resilience (four measured items), which refers to the ability of a firm to operate normally and to recapitalize against risk in a crisis; (2) strategic resilience (four items), which refers to a company's ability to maintain strategic consistency over time, helping it to identify and eliminate adverse factors and to be able to choose the right growth model; (3) relationship resilience (four items), which refers to the reciprocal relationships established between the company and its stakeholders; (4) cultural resilience (four items) refers to the way in which the corporate culture shapes the entrepreneurial spirit of employees and the commitment of employees to the organization; (5) learning resilience (four items), which refers to the ability of the company to cope with the pressures and challenges of learning. Through exploratory factor analysis, the non-conforming measurement items

were eliminated, and the items measuring strategic resilience (SR), capital resilience (CR), relationship resilience (RR), cultural resilience (WR), and learning resilience (LR) were obtained. The Cronbach's alpha coefficient test revealed that the coefficient of each factor of organizational resilience was above 0.8, indicating that this scale has good reliability.

**Table 11.** Exploratory factor analysis.

| Measurement Items | Factor 1 | Factor 2 | Factor 3 | Factor 4 | Factor 5 |
|---|---|---|---|---|---|
| Our business has good cash flow | 0.790 | 0.035 | 0.081 | 0.084 | 0.187 |
| We have multiple sources of financing | 0.896 | 0.067 | 0.073 | 0.097 | 0.198 |
| We have low capital leverage | 0.894 | 0.092 | 0.084 | 0.086 | 0.170 |
| We will make profit maximization the ultimate goal of our business | 0.848 | 0.043 | 0.082 | 0.103 | 0.149 |
| Cronbach's alpha coefficient | | | 0.905 | | |
| Common factor contribution | | | 21.884 | | |
| Our company is able to focus on its core business | 0.044 | 0.816 | 0.005 | 0.014 | 0.201 |
| We pursue a robust strategic growth model | 0.023 | 0.850 | 0.144 | 0.046 | 0.122 |
| We were able to clarify our strategic positioning | 0.080 | 0.804 | 0.091 | 0.042 | 0.155 |
| We are able to balance endogenous and exogenous growth patterns | 0.064 | 0.796 | 0.124 | 0.001 | 0.198 |
| Cronbach's alpha coefficient | | | 0.861 | | |
| Common factor contribution | | | 19.114 | | |
| Our corporate culture is designed to foster a sense of community among our employees | 0.073 | 0.244 | 0.876 | 0.020 | 0.060 |
| Our corporate culture inspires employee morale and spirit | 0.044 | 0.236 | 0.904 | 0.020 | 0.067 |
| Our corporate culture reflects the care and love for our employees | 0.063 | 0.329 | 0.812 | 0.066 | 0.092 |
| Our corporate culture fosters a sense of organizational commitment | 0.070 | 0.325 | 0.786 | 0.009 | 0.110 |
| Cronbach's alpha coefficient | | | 0.920 | | |
| Common factor contribution | | | 11.156 | | |
| We can create unique value for our customers | 0.301 | 0.075 | 0.007 | 0.771 | 0.096 |
| We aim for shared prosperity between companies and stakeholders | 0.294 | 0.118 | 0.022 | 0.836 | 0.107 |
| We have a good reciprocal relationship with our employees | 0.253 | 0.069 | 0.041 | 0.847 | 0.090 |
| We have a good relationship with our investors | 0.247 | 0.083 | 0.041 | 0.838 | 0.084 |
| Cronbach's alpha coefficient | | | 0.900 | | |
| Common factor contribution | | | 12.687 | | |
| We will choose the learning target according to the characteristics of our own company | 0.057 | 0.097 | 0.091 | 0.292 | 0.614 |
| We will have a deep awareness of our situation in time | .079 | 0.242 | 0.104 | 0.194 | 0.785 |
| We will be interested in adjusting our emotions to get into the study state more quickly | 0.088 | 0.192 | 0.091 | 0.145 | 0.801 |
| We will learn more about other experiences to help companies cope with the crisis | 0.087 | 0.209 | 0.081 | 0.143 | 0.826 |
| Cronbach's alpha coefficient | | | 0.826 | | |
| Common factor contribution | | | 16.218 | | |

### 5.2.4. Validation Factor Analysis

In this study, after refining the measurement items of the organizational resilience scale based on exploratory factor analysis, the obtained scale was subjected to validated factor analysis using the LISREL 8.7 software. As seen in Table 12, the latent variable measurement models were $\chi^2/df = 2.01 < 5$, CFI = 0.99 > 0.9, NNFI = 0.98 > 0.9, and RMSEA = 0.044 < 0.1, all of which reached the desired range. Therefore, the model fit of this study is good.

**Table 12.** Validated factor analysis of for the organizational resilience scale.

| Indicators | $\chi^2/df$ | CFI | GFI | AGFI | NNFI | NFI | PGFI | PNFI | RMR | RMSEA |
|---|---|---|---|---|---|---|---|---|---|---|
| Effective range | 2~5 | >0.9 | >0.9 | >0.9 | >0.9 | >0.9 | >0.5 | >0.5 | <0.08 | <0.08 |
| Measured value | 2.01 | 0.99 | 0.94 | 0.92 | 0.98 | 0.97 | 0.72 | 0.82 | 0.034 | 0.044 |

Meanwhile, as shown in Figure 2, the path coefficients of the latent variables corresponding to all the prior variables were above 0.7, and there were significant correlation coefficients among the five latent variables (as shown in Table 13). Thus, the measurement items for organizational resilience developed in this study responded well to the five latent variables and have high reliability and validity.

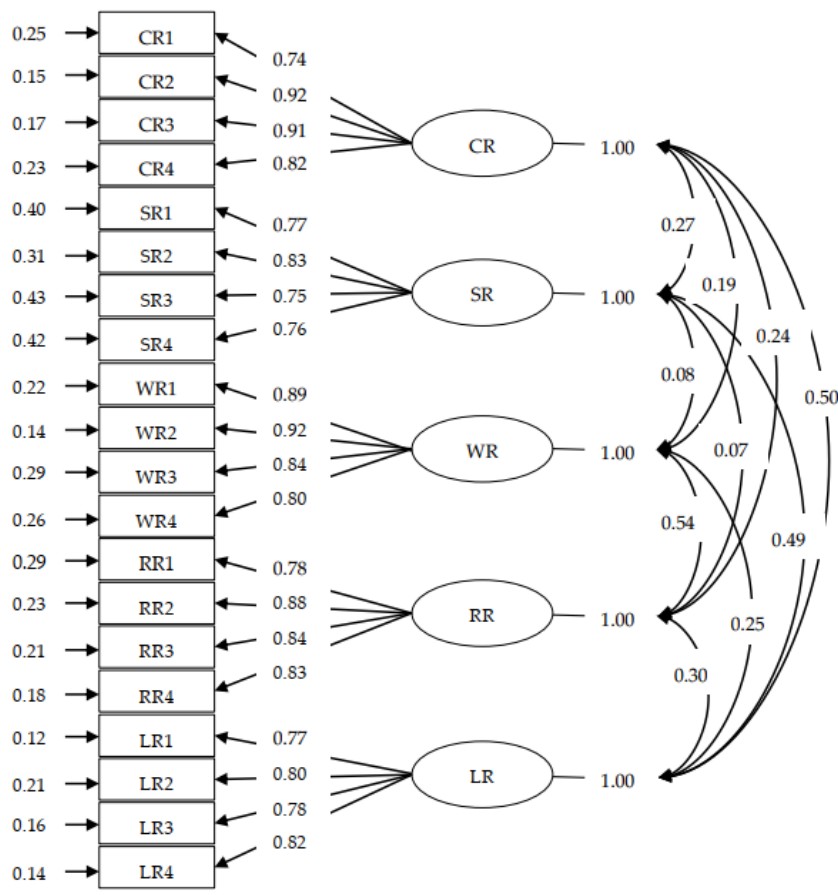

Chi – Square = 322.30, df = 160, P-value = 0.00000, RMSEA = 0.044

**Figure 2.** Organizational resilience validated by factor analysis (CFA). Note: CR stands for capital resilience; SR stands for strategic resilience; WR stands for cultural resilience; RR stands for relationship resilience; and LR stands for learning resilience.

**Table 13.** Correlation coefficient of each variable in descriptive statistics.

| | Average Value | Standard Deviation | 1 | 2 | 3 | 4 | 5 |
|---|---|---|---|---|---|---|---|
| Capital Resilience | 4.72 | 0.903 | 1 | | | | |
| Strategic Resilience | 5.08 | 0.922 | 0.243 ** | 1 | | | |
| Cultural Resilience | 3.41 | 0.950 | 0.180 ** | 0.274 ** | 1 | | |
| Relationship Resilience | 3.52 | 0.904 | 0.23 ** | 0.254 ** | 0.604 ** | 1 | |
| Learning Resilience | 5.17 | 0.858 | 0.446 ** | 0.444 ** | 0.233 ** | 0.259 ** | 1 |

Note: ** indicates $p < 0.01$.

## 6. Conclusions and Discussion

### 6.1. Conclusions

With the development of the digital economy, the environment in which organizations operate is increasingly volatile. Survival and growth in a dynamic environment are central goals for organizations. Therefore, the importance of organizational resilience is recognized by both scholars and practitioners. This study analyzed six highly resilient companies, namely Southwest Airlines, Apple, Microsoft, Starbucks, Kyocera, and Lego. We provided an in-depth study of the dimensions and measurement components of organizational resilience. We have also clarified the process of organizational response to crises (as shown in Figure 3). First, this study clarified the concept of organizational resilience. Specifically, organizational resilience refers to the ability of an organization to reconfigure organizational resources, optimize organizational processes, reshape organizational relationships in a crisis, recover quickly from the crisis, and use the crisis to achieve counter-trend growth. Second, we specified five dimensions of organizational resilience: capital resilience, strategic resilience, relationship resilience, cultural resilience, and relationship resilience. Capital resilience refers to a company's ability to operate normally and to recapitalize against risk in a crisis. Strategic resilience refers to a company's ability to maintain strategic consistency over time, helping it to identify and eliminate adverse factors and to be able to choose an appropriate growth model. Relational resilience refers to the reciprocal relationships established between a company and its stakeholders. Cultural resilience refers to the way in which the corporate culture shapes the entrepreneurial spirit of employees and the commitment of the employees to the organization. Learning resilience refers to a company's ability to cope with the stresses and challenges of learning. Finally, this study underwent a process of exploratory factor analysis and validation factor analysis to develop a 20-item scale for measuring organizational resilience.

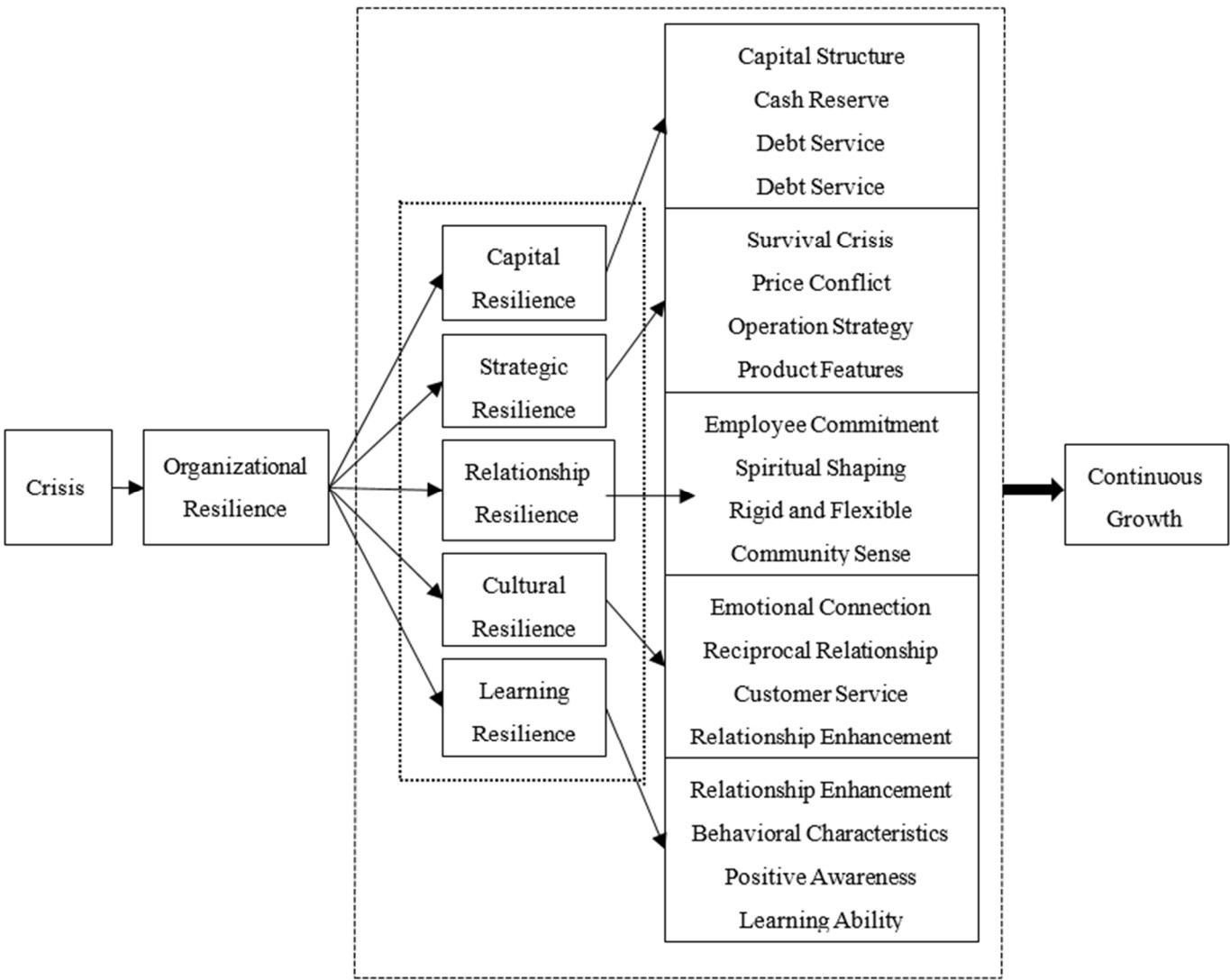

**Figure 3.** The organizational resilience process.

### 6.2. Discussion and Practical Implications

Compared to previous studies that measured organizational resilience, this study pioneered the use of a multi-case analysis approach. We drew on procedural rooting theory to obtain five dimensions of organizational resilience, which both complement and build on existing research. Previous studies found that organizational resilience is a multidimensional construct [27,41,42], which our study also validated. Scholars have argued that organizational resilience contains organizational learning and organizational resource dimensions [37,38,44]. Our study found that organizational resilience contains learning resilience and capital resilience. Learning plays an important role in organizations' response to crises by enhancing their organizational capabilities [101]. Moreover, organizational resources can reduce organizational vulnerability and increase organizational resistance to the effects of crises [43]. It has also been noted that the idle resources that organizations have play an important role in providing flexibility and enhancing the organization's ability to cope with crises [102–104]. Thus, learning resilience and capital resilience are important components of organizational resilience. Scholars have focused on the important role of relationships in organizational resilience [40,86,87]. Our study found that organizational resilience encompasses relationship resilience, which corroborates these studies' findings. We also found that organizational resilience also includes strategic resilience and cultural

resilience by conducting case studies of companies such as Southwest Airlines, which is a useful supplement to existing organizational resilience research.

A practical implication of this study is that building organizational resilience and accumulating corporate resilience assets does not happen overnight, but requires both time and a long-term strategic design, detailed planning, and effective measures. First, we need to implement a lean strategy to build strategic resilience. On the one hand, it is important for companies to have an ambitious vision and mission. The vision and mission of a company play an important role in its strategy and goals, and a common vision facilitates the growth of the company and paves the way for its future development. On the other hand, it is important for companies to set clear development goals and shape core competencies to match them. An ambitious vision and corporate mission help guide employees to focus on long-term development, while clear development goals help motivate organizational members and promote efficient synergy and mutual support within the organization. The matching of organizational members' competencies with organizational goals helps to discover the impact of organizational members' competencies on organizational goals.

Second, companies need to practice sound capital to shape capital resilience. Companies need to maintain adequate cash flow. Crises are unpredictable, so companies must be prepared before they arrive, and they need to determine the appropriate cash reserves based on the use of working capital. It is also necessary to control the level of capital leverage of the company. In the event of a crisis, companies with high capital leverage often find it difficult to cope with crises. It is necessary for them to adopt a sound financial strategy in the development process and keep their capital leverage within a reasonable range.

Third, it is important to practice reciprocal relationships to build relationship resilience. Higher relational resilience helps companies form strong cohesion with employees, customers, and investors when a crisis hits, thus, helping companies emerge from the crisis. Moreover, we want to take employees as the main body, so that employees and enterprises constitute a community of interests. Employees are one of the most important resources of the enterprise and the main body of enterprise value creation. Therefore, cultivating the organizational loyalty of employees and retaining them helps the enterprise enhance the dedication and cohesion of employees. As customers are the cornerstone of enterprise development, establishing a strong relationship between enterprises and customers can help enterprises overcome crises.

Fourth, we need to implement a culture of excellence to build cultural resilience. A resilient organizational culture is conducive to shaping a sense of community among organizational members, which can help organizations survive crises. At the same time, a relaxed organizational climate is more conducive to good organizational performance [105]. On the one hand, we should focus on caring and happiness. Caring for employees is good for employees to experience the feeling of home, which will motivate them to work harder to help the company cope with crises. Happy experiences can make the organization members have passion and can enhance the efficiency of the organization members. On the other hand, it is important to shape the commitment of organization members. Committed employees are conducive to the harmonious and stable development of the company.

However, as an exploratory study, this study still has some shortcomings. First, organizational resilience is a multidimensional structure, and the differences in industries may give different characteristics to organizational resilience. The samples selected for this study are mainly from the fields of technology, service, and manufacturing. Therefore, the structure of organizational resilience in other industries still needs to be further explored. Second, although Chinese scholars have measured organizational resilience, their research results have been integrated on the basis of existing studies. Fewer studies have explored the structure of organizational resilience from the perspective of multiple case studies. Although this current study has developed a scale for measuring organizational resilience, the crises faced by companies are constantly changing with the changes in the business environment and with the development of digital technology. The structure and

measurement of organizational resilience may become more enriched and mature in the future as the external environment changes.

**Author Contributions:** Conceptualization, R.C. and Y.X.; methodology, Y.L.; validation, Y.L. and R.C.; formal analysis, Y.X.; investigation, R.C.; resources, R.C. and Y.X.; data curation, R.C.; writing—Original draft preparation, R.C.; writing—Review and editing, Y.L.; visualization, Y.X.; supervision, Y.L. and R.C. All authors have read and agreed to the published version of the manuscript.

**Funding:** This research was funded by the National Social Science Fund of China, grant number 16AGL004; 2019 Youth Project of Fujian Province Social Science Planning, grant number FJ2019C031.

**Institutional Review Board Statement:** Not applicable.

**Informed Consent Statement:** Not applicable.

**Data Availability Statement:** Not applicable.

**Acknowledgments:** The authors would like to thank the anonymous reviewers for their reviews and comments.

**Conflicts of Interest:** The authors declare no conflict of interest.

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
