# Peer review of "Defining, Conceptualizing, and Measuring Organizational Resilience: A Multiple Case Study"

_sustainability, doi:10.3390/su13052517_

Round 1
Author Response
Dear Reviewers: First of all, I would like to thank the editors and reviewers for their review of this paper and their valuable comments. Combined with the revisions, the authors have made the following revisions, which are detailed in the revision notes and the revised draft. Point 1: The paper seems to me conceptually interesting. It would be very interesting to have some ideas concerning the types of qualitative or quantitative evaluation of the proposed scale items. Also it would be interesting to have some elements on the preponderance of such or such scale items in the non-resilience of companies that will not have been. Response 1: Thank you very much for your valuable opinions. According to this revision, the purpose of this study is to obtain the structure of organizational resilience through the analysis of the cases, and on this basis, the development of the organizational resilience scale. Therefore, in order to ensure the reliability and validity of the scale, the internal consistency and structural validity, content validity, criterion validity, and face validity of the scale were examined in this study, and the results showed that the scale has good reliability and validity. Please see lines 405 to 457 in the paper for more details. In the authors' choice of cases, Starbucks Corporation was once on the verge of collapse because of a lost strategy, and later led Starbucks out of the crisis because the business leaders redeveloped their strategic model. In addition, Lego was once in an existential crisis due to losses, and then the business leaders developed a new strategic transformation route, re-programmed resources, and finally came out of the capital crisis. The experiences of these companies provide evidence of the effectiveness of the organizational resilience structure summarized in this study. Therefore, this study is equally applicable in firms that are not resilient.
Reviewer 2 Report
Dear authors,
I would like to complement your for preparing a high-quality manuscript. It will certainly add the value to the field. However, I would also like to draw you attention to some remaining issues. Please find below few comments that need to be addressed:
(1) Please note briefly in your abstract what is your sample size for scale validation.
(2) Please check and report about interrater agreement (p. 8).
(3) Please describe and explain more thoroughly how you checked for different types of measurement validity (i.e. construct, content, face, and criterion). This is essential for making your scale more rigor.
Wish you all the best in further paper development.
Author Response
Dear Reviewers:
First of all, I would like to thank the editors and reviewers for their review of this paper and their valuable comments. Combined with the revisions, the authors have made the following revisions, which are detailed in the revision notes and the revised draft.
Point 1: Please note briefly in your abstract what is your sample size for scale validation.
Response 1: Thank you very much for your valuable opinions. In this study, 900 questionnaires were distributed and 723 questionnaires were collected, and invalid questionnaires were eliminated based on the criteria of whether the questionnaires were filled out completely and whether the questionnaires were filled out randomly, and finally 526 valid questionnaires were obtained. Therefore, according to this revision, in line 59 of the abstract we described the number of valid questionnaires collected as follows: Specifically, we conducted exploratory factor analysis and validation factor analysis on the 526 valid data collected.
Point 2: Please check and report about interrater agreement (p. 8).
Response 2: Thank you very much for your valuable opinions. According to the revision, the authors used the Intraclass Correlation Coefficient (ICC) measure to test the internal consistency of the scale of organizational resilience, drawing on Landers' (2015) requirements for internal consistency testing, and the results showed that this scale has good internal consistency. See rows 405 to 414 for more details.
Point 3: Please describe and explain more thoroughly how you checked for different types of measurement validity (i.e. construct, content, face, and criterion). This is essential for making your scale more rigor.
Response 3: Thank you very much for your valuable opinions. Based on the modifications, the authors conducted a validity analysis of the scale. First, the authors analyzed the structural validity by combining Chan and Idris' (2017) study, which used the measure KMO and Bartlett's Test of Sphericity as indicators of structural validity, and found that there was sufficient evidence to conclude that the questionnaire was valid at the 99% significance level. Second, the authors analyzed the content validity by using a 4-point expert scale, with scores from 1 to 4 representing "not relevant", "weakly relevant", "more relevant", and "very relevant", respectively. In this study, six experts in the field were invited to rate the items of the scale, and the results of both I-CVI and S-CVI were 1. Therefore, the content validity of the scale was good. Third, the authors analyzed the criterion validity, and the results showed that the scale had good criterion validity. Fourth, the authors analyzed the face validity. In this study, two professors and five doctoral students in this field were invited to evaluate the face validity of this scale based on the preliminary research, and finally, the experts unanimously agreed that this scale has good face validity. For details, see rows 415 to 457.

Reviewer 3 Report
Manuscript ID: sustainability-1111877
Title: "Defining, conceptualizing, and measuring organizational resilience: A multiple case study"
REVIEW REPORT
In the paper, the Authors used an exploratory case study approach to explore the process of organizational resilience among six highly resilient companies (South-west Airlines, Apple, Microsoft, Starbucks, Kyocera, and Lego).
The topic of the paper is very interesting as well as the academic contribution of the work. The paper is clear, well written and well organized. The methodological approach is technically correct. The references incluse demonstrate a good knowledge of the field by Authors, and literature review contributes to the definition of the concrete contribution of the paper. English language and style are fine.
Only a minor revision: the following study should be considered.
https://doi.org/10.3390/su12083340
Author Response
Dear Reviewers:
First of all, I would like to thank the editors and reviewers for their review of this paper and their valuable comments. Combined with the revisions, the authors have made the following revisions, which are detailed in the revision notes and the revised draft.
Point 1: In the paper, the Authors used an exploratory case study approach to explore the process of organizational resilience among six highly resilient companies (South-west Airlines, Apple, Microsoft, Starbucks, Kyocera, and Lego). The topic of the paper is very interesting as well as the academic contribution of the work. The paper is clear, well written and well organized. The methodological approach is technically correct. The references incluse demonstrate a good knowledge of the field by Authors, and literature review contributes to the definition of the concrete contribution of the paper. English language and style are fine. Only a minor revision: the following study should be considered. https://doi.org/10.3390/su12083340
Response 1: Thank you very much for your valuable opinions. According to the revision, the author carefully read and analyzed the paper "evaluating climate between working excellence and organizational innovation: what comes" At the same time, in line 593 of this study, it points out that loose organizational atmosphere is more conducive to the organization to achieve good performance, which enriches the research results of this study.
